# Metabolic Activation of CsgD in the Regulation of *Salmonella* Biofilms

**DOI:** 10.3390/microorganisms8070964

**Published:** 2020-06-27

**Authors:** Akosiererem S. Sokaribo, Elizabeth G. Hansen, Madeline McCarthy, Taseen S. Desin, Landon L. Waldner, Keith D. MacKenzie, George Mutwiri, Nancy J. Herman, Dakoda J. Herman, Yejun Wang, Aaron P. White

**Affiliations:** 1Vaccine and Infectious Disease Organization-International Vaccine Centre, University of Saskatchewan, Saskatoon, SK S7N 5E3, Canada; ats172@mail.usask.ca (A.S.S.); egh367@mail.usask.ca (E.G.H.); mcm876@mail.usask.ca (M.M.); landonwaldner@hotmail.com (L.L.W.); gmm777@mail.usask.ca (G.M.J.); nfairburn@msn.com (N.J.H.); dakoda.herman@mail.utoronto.ca (D.J.H.); 2Department of Biochemistry, Microbiology and Immunology, University of Saskatchewan, Saskatoon, SK S7N 5E5, Canada; taseen.desin@usask.ca; 3Basic Sciences Department, King Saud bin Abdulaziz University for Health Sciences, Riyadh 11481, Saudi Arabia; 4Institute for Microbial Systems and Society, Faculty of Science, University of Regina, Regina, SK S4S 0A2, Canada; keith.mackenzie@uregina.ca; 5Department of Biology, University of Regina, Regina, SK S4S 0A2, Canada; 6Department of Cell Biology and Genetics, School of Basic Medicine, Shenzhen University Health Science, Shenzhen 518060, China; wangyj@szu.edu.cn

**Keywords:** biofilm, *Salmonella*, CsgD, curli, cellulose, CpxR

## Abstract

Among human food-borne pathogens, gastroenteritis-causing *Salmonella* strains have the most real-world impact. Like all pathogens, their success relies on efficient transmission. Biofilm formation, a specialized physiology characterized by multicellular aggregation and persistence, is proposed to play an important role in the *Salmonella* transmission cycle. In this manuscript, we used luciferase reporters to examine the expression of *csgD*, which encodes the master biofilm regulator. We observed that the CsgD-regulated biofilm system responds differently to regulatory inputs once it is activated. Notably, the CsgD system became unresponsive to repression by Cpx and H-NS in high osmolarity conditions and less responsive to the addition of amino acids. Temperature-mediated regulation of *csgD* on agar was altered by intracellular levels of RpoS and cyclic-di-GMP. In contrast, the addition of glucose repressed CsgD biofilms seemingly independent of other signals. Understanding the fine-tuned regulation of *csgD* can help us to piece together how regulation occurs in natural environments, knowing that all *Salmonella* strains face strong selection pressures both within and outside their hosts. Ultimately, we can use this information to better control *Salmonella* and develop strategies to break the transmission cycle.

## 1. Introduction

*Salmonella enterica* strains that cause gastroenteritis and typhoid fever were recently ranked first and second in terms of global disease impact (i.e., disability adjusted life years) among 22 of the most common food borne pathogens [1]. *S. enterica* strains are distributed within >2000 serovars, with yearly estimates of approximately 94 million cases of gastroenteritis [2] and 21 million cases of typhoid fever worldwide [3]. The serovars associated with typhoid fever (i.e., Typhi, Paratyphi and few others [4]) consist of human-restricted strains and are collectively referred to as typhoidal *Salmonella* (TS). The serovars associated with gastroenteritis (i.e., Typhimurium, Enteritidis and >1600 others) consist of host-generalist strains and are collectively referred to as nontyphoidal *Salmonella* (NTS) [5]. NTS outbreaks are relatively common occurrences and are often linked to the consumption of contaminated food produce, such as poultry [6,7], fruits, vegetables [8,9] and processed foods [10]. In general, NTS strains have a remarkable ability to persist and survive in harsh conditions, including extremes of drying and nutrient limitation [11,12,13].

The majority of NTS strains can form biofilms, a specialized physiology that is characterized by multicellular aggregation, long-term survival, and resistance. Biofilm formation has been linked to *Salmonella* persistence on food surfaces, plants, and other produce, and is thought to provide protection during food processing [14,15,16]. Aside from the food-borne aspects, biofilm formation is hypothesized to be an integral part of the life cycle of gastroenteritis-causing *Salmonella* strains, by ensuring long-term survival of cells as they cycle between hosts and the environment [14,17]. We have speculated that biofilms are connected to the host generalist lifestyle since the environment (soil and water) would be a common collecting point for multiple host species. In contrast, there is widespread loss of biofilm formation in TS strains and other more invasive strains, such as the specialized NTS strains associated with human bloodstream infections in sub-Saharan Africa [18,19], although TS produce alternative biofilms on gallstones inside human carriers [20]. There are multiple selection pressures acting on biofilm formation in diverse *Salmonella* strains. In short, biofilms are thought to represent the most dominant form of bacterial life on the planet and understanding the regulation of this specialized physiology is important.

Biofilm-forming strains of *S. enterica* can be identified by the production of distinct rdar (red, dry and rough) morphotype colonies when grown on agar-containing media supplemented with the dye Congo red. Cells within the colony are held together by curli fimbriae for short-range interactions and cellulose for long-range interactions [12,21,22]. In addition, other polymers are part of the extracellular biofilm matrix, including polysaccharides (i.e., O-Ag capsule, colanic acid and cellulose) proteins (i.e., BapA, curli, and flagella), lipopolysaccharides and DNA [8,23]. Curli, cellulose and the biofilm matrix impart survival and persistence properties on cells within the biofilm [12,13,14]. It is not known if the survival traits are specific to the polymers themselves or are emergent properties associated with cells entering a unique physiological state [24,25,26]. Perhaps the microenvironments generated within a biofilm are responsible for the adaptations, heterogeneity and cellular differentiation observed during biofilm formation [27,28].

In *Salmonella*, regulation of curli, cellulose and other polymers is coordinated through CsgD, the main transcriptional controller of biofilms. The activation of CsgD in vitro has been well-defined, with growth conditions of low osmolarity, lower temperatures and limiting nutrients necessary to activate *csgD* transcription [29]. Expression of *csgD* is repressed tightly at early stages of growth but is induced up to 370-fold when cells enter the stationary phase of growth [30]. The same general principles apply in *E. coli*, which shares the CsgD, curli, and cellulose biofilm components [31]. In the stationary phase of growth, cell density in the culture is high, nutrients become limiting and cells express the alternative sigma factor RpoS [32]. RpoS controls the general stress response [33] and selectively transcribes *csgD* [34,35]. The effects of osmolarity are mediated through the EnvZ/OmpR and CpxA/CpxR two-component signal transduction systems [36]. In low osmolarity, low levels of phosphorylated OmpR bind to a high-affinity binding site −50.5 bp upstream of the *csgD* transcription start sites, which activates *csgD* transcription [37]. In high osmolarity, transcription is repressed through binding of phosphorylated CpxR to multiple sites on the *csgD* promoter [36], as well as phosphorylated OmpR binding to a low-affinity site in the *csgD* promoter [38]. We realized that the complex regulatory network behind *csgD* activation [8] was even more dynamic when it was discovered that CsgD was produced in a bistable manner [39,40,41]. Biofilm cells are maintained in a CsgD-ON state due to a predicted feed-forward loop consisting of RpoS, CsgD and IraP, a protein that stabilizes RpoS [35,42]. The remaining single cells are in a CsgD-OFF state and express several important virulence factors [14]. The connection between persistence and virulence during biofilm formation brings into question the hierarchical regulation of this process, as well as determining how individual cells become activated and remain in their CsgD-ON or -OFF states.

The regulation of *Salmonella* biofilms is also strongly influenced by the intracellular levels of the second messenger, cyclic-di-GMP (c-di-GMP). It is synthesized from two guanosine 5′-triphosphate molecules by diguanylate cyclases (DGCs) and degraded by specific phosphodiesterases (PDEs). In general, high levels of c-di-GMP are associated with biofilm formation, sessility and persistence, and low levels of c-di-GMP are associated with motility and virulence [43]. The change in c-di-GMP levels in *S. enterica* is controlled by the enzymatic activity of 17 different DGCs and PDEs. For biofilms, the cellulose synthase enzyme, BcsA, is allosterically activated by c-di-GMP that is produced by AdrA, a DGC that is transcriptionally activated by CsgD. Expression of CsgD itself is influenced by c-di-GMP synthesis and breakdown by a network of DGC and PDE enzymes [44]. The importance of c-di-GMP regulation is underscored by the observation that *S. enterica* isolates that are defective in the production of DGCs are both avirulent and unable to form biofilms [45].

In this manuscript, we analyzed the regulation of *csgD* transcription and the activity of CsgD through activation of curli biosynthesis (*csgBAC*) and cellulose production (*adrA*). We examined the response to different environmental signals (i.e., temperature, osmolarity, nutrients) and discovered that there is a hierarchy of regulation. These environmental signals were selected because their effects on csgD expression before induction have been well established and we hypothesize that these conditions would be encountered during food processing and in both host and non-host environments. We established that the CsgD system responds differently or not at all to known regulatory inputs once it has been activated. This is similar to some dedicated, point of no return processes, such as sporulation in *Bacillus subtilis* [46]; however, we show that the CsgD system can be reversed by other signals, such as glucose. The implications for the *Salmonella* lifecycle are discussed.

## 2. Materials and Methods

### 2.1. Bacterial Strains, Media, and Growth Conditions

The bacterial strains used in this study are listed in Table 1. For standard growth, strains were inoculated from frozen stocks onto LB agar (lysogeny broth, 1% NaCl, 1.5% agar) supplemented with appropriate antibiotic (50 µg mL^−1^ kanamycin (Kan), or 5 µg mL^−1^ tetracycline (Tet)) and grown overnight at 37 °C. Isolated colonies were used to inoculate 5 mL LB broth and the culture was incubated for 18 h at 37 °C with agitation at 200 RPM.

For analysis of colony morphology and gene expression, 4 µl of overnight culture was spotted on 1% tryptone agar supplemented with 0.2% freshly made glucose, 25 mM salt or 100 mM salt (agar supplemented with glucose were used within 24 h). Plates were incubated at 28 °C or 37 °C for two days. Visible and luminescence images were captured with a spectrum CT in vivo imaging system (PerkinElmer, Waltham, MA, USA).

### 2.2. Generation of *S. typhimurium* 14028 Mutant Strains

Lambda red recombination [49] was used to generate ∆*cpxR* and ∆*iraP S.* Typhimurium mutant strains. Primers containing 50-nuclelotide sequences on either side of *cpxR* or *iraP* (Table 2) were used to amplify the *cat* gene from pKD3 using Phusion high-fidelity DNA polymerase (New England Bio-Labs, Ipswich, MA, USA). The PCR products were solution purified and electroporated into *S*. Typhimurium 14028 cells containing pKD46. Mutants were first selected by growth at 37 °C on LB agar supplemented with 10 µg ml^−1^ chloramphenicol (Cam) before streaking onto LB agar containing 34 µg ml^−1^ Cam. PCR primers upstream and downstream of *cpxR* or *iraP* (Table 2) were used to amplify sequence from the genome of mutant *S. typhimurium* 14028 strains and verify loss of the corresponding open reading frames. The ΔcpxR or Δ*iraP* mutations were moved into a clean *S. typhimurium* strain background with P22 phage [50]. The *cat* gene was resolved from the chromosome using pCP20 [49].

### 2.3. Generation of Bacterial Luciferase Reporters and Other Plasmid Vectors

Luciferase fusion reporter plasmids containing the promoters of *csgDEFG, csgBAC* and *adrA* have been previously described [12]. The *cpxP* reporter plasmid was generated to monitor the levels of CpxA/CpxR activation within the cell. The intergenic region containing the *cpxR* and *cpxP* promoter sequences was PCR amplified from *S*. Typhimurium 14028 using primers cpxR1 and cpxR2 (Table 2) and Phusion high-fidelity DNA polymerase (New England BioLabs, Ipswich, MA, USA). The resulting PCR product was purified, sequentially digested with *Xho*I and *Bam*HI, and ligated (in the *cpxP* direction) using T4 DNA ligase (New England BioLabs, Ipswich, MA, USA) into pU220 digested with *Xho*I and *Bam*HI. The *stm1987* luciferase reporter plasmid was generated similarly using primers STM14_2408for1 and STM14_2408rev2 (Table 2), with cloning into pCS26. PCR screening with primers pZE05 and pZE06 was used to verify the successful fusion of promoter regions to *luxCDABE.*

For plasmid-based overexpression of cyclic-di-GMP related enzymes, fragments containing *stm1987* and *yhjH* genes with their native promoters were PCR amplified from *S. typhimurium* 14028 gDNA using Phusion high-fidelity DNA polymerase and appropriate primers (Table 2). Resulting PCR products were purified, digested with *Eco*RI and *Aat*II, and ligated using T4 DNA ligase into *Eco*RI/*Aat*II-digested pBR322. The pACYC-*rpoS* plasmid vector has previously been described [48]. Reporter plasmids and overexpression plasmids were co-transformed into *S. typhimurium* strains by electroporation and selected by growth at 37 °C on LB agar supplemented with 50 µg mL^−1^ Kan (pCS26) and 10 µg mL^−1^ Tet (pBR322 or pACYC).

### 2.4. Luciferase Reporter Assays

96-well bioluminescence assays were performed with *S*. Typhimurium luciferase reporter strains. Overnight cultures were diluted 1 in 600 into individual wells of black, clear bottom 96-well plates (9520 Costar; Corning Life Sciences, Tewksbury, MA, USA) containing 150 µL of 1% tryptone broth supplemented with 50 µg mL^−1^ of kanamycin (Kan). When noted, media was supplemented before growth with NaCl (25,150 mM), sucrose (50,150 mM), CuCl_2_ (1 mM), casamino acids (12%) or individual amino acids (15 mM) to the final concentrations as indicated. For the addition of media supplements during growth, cells were inoculated into 135 µL of media and grown for 18 h at 28 °C before supplements were added as 15 µL aliquots to the appropriate wells. This included glucose ranging from 25–150 mM. To minimize evaporation of the media during the assays, cultures were overlaid with 50 µL of mineral oil per well. Cultures were assayed for absorbance (600 nm, 0.1 s) and luminescence (1s; in counts per second (CPS)) every 30 min during growth at 28 °C with agitation in a Victor X3 multilabel plate reader (Perkin-Elmer, Waltham, MA, USA).

## 3. Results

### 3.1. Osmolarity Has No Effect Once csgD Transcription Is Activated

Osmolarity is a key regulatory factor for Salmonella biofilm formation in vitro [29,51]. In the presence of high concentrations of NaCl, *csgD* transcription is abolished [52]. In *E. coli*, this repression is mediated through the CpxA/R two-component system [37]. We performed transcription experiments with *S. enterica* serovar Typhimurium ATCC 14028 (i.e., *S. typhimurium* 14028). Consistent with *E. coli*, expression of *csgD* was highest in low osmolarity media (i.e., no salt) and reduced sequentially in media supplemented with increasing concentrations of NaCl (Figure 1A). Reduced *csgD* expression in media supplemented with 75 mM or more salt correlated with basal expression of *csgBAC* (curli biosynthesis) and *adrA* (cellulose biosynthesis) (Figure 1B,C). To gauge the activity of the CpxA/R system, and its potential role in repression, we monitored expression of *cpxP*, a known regulatory target of CpxR [53]. Expression of cpxP was highest in media supplemented with 150 mM NaCl (Figure 1D), which was inversely correlated with *csgD* expression levels. This was consistent with CpxR-mediated repression of *csgD* transcription. 

Regulation of *csgD* expression via the CpxA/R system is thought to be a dynamic process involving surface-sensing and feedback during curli production [36,54]. Therefore, we performed experiments where salt was added to growing cultures after 18 h of growth, rather than being premixed into the media before growth. At 18 h of growth, *csgD* expression level is rapidly increasing and *csgBAC* and *adrA* expression are just beginning to increase [26]. Under these conditions, *csgD* expression did not change when increasing concentrations of salt were added during growth; the expression curves were nearly superimposable regardless of the amount of salt added (Figure 1E). Expression of *csgBAC* was also not inhibited by the addition of salt and was actually increased at high salt concentrations (Figure 1F). For *adrA*, mild repression was observed, but expression was well above background levels, even in the presence of 150 mM salt (Figure 1G). *cpxP* expression, on the other hand, was similar to the premixed experiments, with highest expression in the 150 mM salt media and lowest expression in non-supplemented media (Figure 1H). These results indicated that the Cpx system was activated by the addition of salt during growth but was no longer causing repression of *csgD* transcription and the downstream genes involved in curli and cellulose production. 

### 3.2. Repressive Effect of CpxR on csgD Transcription Is Alleviated During Growth

To examine the effects of Cpx-mediated repression of *csgD* transcription in more detail, we monitored gene expression in a Δ*cpxR* mutant background. The Cpx system can be activated by high concentrations of metals and a variety of other signals, with each thought to represent a form of periplasmic stress [54,55]. Growth of *S. typhimurium* 14028 in media supplemented with 1 mM copper chloride resulted in activation of the Cpx system, as measured by an increase in *cpxP* expression (Figure 2A, + inducer). As expected, *cpxP* expression was off in the Δ*cpxR* strain background (Figure 2A, red line). In the presence of copper chloride, *csgD* expression reached high levels, similarly to when in the presence of non-supplemented media (Figure 2B). There was also a slight increase in the Δ*cpxR* strain, which was consistent with CpxR being a repressor of *csgD* transcription. This effect was more pronounced for *csgBAC*, as expression was approximately four times higher in the Δ*cpxR* strain (Figure 2C). We performed the same experiment with the addition of copper chloride after 18 h of growth. The Cpx system was activated normally, as shown by elevated *cpxP* expression levels in the presence of the inducer (Figure 2D). However, expression of *csgDEFG* and *csgBAC* was unchanged in the Δ*cpxR* mutant strain, showing no evidence of CpxR-mediated repression (Figure 2E,F). This indicated that once the *csgD* network was activated, the system was unresponsive to CpxR.

We also measured biofilm gene expression after the addition of sucrose (Figure 3). In *E. coli*, sucrose has been shown to repress *csgD* transcription, due to the activity of H-NS [36]. Sucrose is also a cleaner measure of osmolarity because unlike salt, it does not result in a change of ionic strength. In general, the *csgDEFG*, *csgBAC* and *adrA* expression profiles were consistent with what was measured in response to salt addition. When sucrose was added to the media before growth, significant repression was observed for all three promoters (Figure 3A–C). However, when sucrose was added to growing cultures at 18 h, there was no repression measured (Figure 3E–G). The addition of sucrose had minimal effect on *cpxP* expression (Figure 3D,H), and, therefore, did not appear to engage the CpxR/A system, similar to what was observed in *E. coli* [36]. These results indicated that the *S. typhimurium*
*csgD* biofilm network is not influenced by changes in osmolarity after it has been activated. Moreover, this appears to be a general effect that is not restricted to repression by the Cpx system.

### 3.3. Temperature and Glucose Repress csgD Expression

The idea that the biofilm system can become unresponsive to known regulatory inputs once it is activated fits with one of the hallmarks of a bistable gene expression system, in that a proportion of cells can remain activated even when the inducer is absent [56]. There are other bacterial physiologies, such as sporulation, where the cellular differentiation process is irreversible [46]. The *csgD* biofilm network has been shown to have bistable expression [40,42]. We wondered if the response we had observed with osmolarity was representative of a non-reversible system.

In most *Salmonella* and *E. coli* strains, *csgD* expression and biofilm formation is activated at temperatures below 30 °C and repressed at higher temperatures [52]. There are strains that produce biofilms at higher temperatures (i.e., 37 °C), but these typically possess single nucleotide polymorphisms in the *csgD* promoter region that allows for disregulated expression [50,52,57,58]. We tested whether increased temperature could shut off activated biofilm gene expression by first growing cells at 28 °C for 18 h and then shifting the temperature to 30, 32, 35, or 37 °C. At 30 °C or 32 °C, there was a measurable drop in *csgDEFG*, *csgBAC* and *adrA* expression, but it was still above background levels (Figure 4A–C). However, a temperature shift above 32 °C reduced gene expression to baseline levels (Figure 4A–C). This showed that high temperature was able to override the activation of *csgD* and biofilm related genes.

Glucose is another powerful repressor of *csgD* expression and biofilm formation in vitro [26,59]. For *S. typhimurium*, the addition of glucose to growing cultures at 18 h rapidly abolished *csgDEFG* (Figure 4D), *csgBAC* (Figure 4E) and *adrA* (Figure 4F) expression, even at the lowest added concentration of 25 mM. We tested lower concentrations of glucose and found that in each case, *csgD* transcription was immediately repressed but was restored at later timepoints, presumably when all glucose was metabolized. This showed that glucose was a powerful repressive signal. Together, these experiments showed that activation of the *S*. Typhimurium *csgD* biofilm network is a reversible process and suggested the existence of a regulatory hierarchy.

### 3.4. Effect of Casamino Acids on Biofilm Formation

Expression of *csgD* is known to be activated once cells reach a critical density and nutrients start to run out [30]. Since 1% tryptone is primarily an amino acid-based media [60], we speculated that the addition of amino acids would reduce or delay expression of *csgD* and other biofilm genes. Casamino acids (CAA) are a complex mixture of amino acids and small peptides that are used for nutritional investigations of bacterial growth. The addition of CAA to the medium prior to *S*. Typhimurium growth reduced *csgDEFG* expression approximately 15-fold in the presence of 0.5, 1.0 or 2.0% CAA (Figure 5A). *csgBAC* and *adrA* expression dropped to near baseline levels when CAA was added at the beginning of growth (Figure 5B,C). The addition of CAA to growing *S. typhimurium* 14028 cultures also reduced the expression of all three promoters, but in a more dose-dependent manner. Expression of *csgDEFG* was reduced to ~75%, 50% and 25% of initial levels after the addition of 0.5% CAA, 1.0% CAA and 2% CAA, respectively (Figure 5D). Expression of *csgBAC* was reduced after the addition of 0.5% or 1.0% CAA, but the promoter was still considered active, whereas expression returned to baseline after the addition of 2.0% CAA (Figure 5E). *adrA* expression returned to near baseline levels, even with the addition of 0.5% CAA (Figure 5F). These experiments demonstrated that there is a metabolic feedback into *csgD* expression and that the system responds differently once it has been activated.

### 3.5. Differing Effects of Individual Amino Acids on csgD Gene Expression

We wanted to test how individual amino acids contributed to the repression of biofilm gene expression caused by CAA. We measured *csgBAC* expression i.e., curli production) as a proxy for biofilm formation and as readout for CsgD activity. Only Asn, Pro and Arg had a direct repressive effect on *csgBAC* expression when added individually (Figure 6A; blue bars). The expression curves were lower for the entirety of growth (Figure 6B). Six amino acids had no significant effect (Figure 6A, grey bars; examples in 6C) and seven amino acids caused an increase in expression (Figure 6A, pink bars). The addition of Gly and Thr yielded an approximately three-fold boost to *csgBAC* expression (Figure 6D), which was unexpected. These results indicated that the repression caused by CAA must have been due to the cumulative effect of multiple amino acids.

When individual amino acids were added to *S*. Typhimurium cultures after 18 h of growth, the effects on *csgBAC* expression were not predictable based on their previous groupings (Figure 6E; see color distribution). No amino acids caused a decrease in expression, and some that were repressive when added before growth (i.e., Arg, Pro), now caused a significant boost in expression (Figure 6E,F). Eight amino acids had no signficant difference from the water control (Figure 6G, Lys, Ser). Val, Ala, Gln and Thr led to increased *csgBAC* expression when added before or during growth, suggesting that these amino acids have a positive effect on curli fimbriae synthesis. Glycine, on the other hand, had no significant effect when added during growth (Figure 6H). Overall, we could not explain the differing effects of individual amino acids when added during growth. However, the results were consistent with our previous observation that the *S. typhimurium* biofilm network responds differently to regulatory inputs after the *csgD* network has been activated.

### 3.6. Regulation of Rdar Morphotype on Agar-Containing Media

In the bistable expression of CsgD, the proportion of cells in the “ON” state is thought to be maintained by a feed-forward loop consisting of RpoS, the stationary phase sigma factor that controls *csgD* transcription, IraP, a protein that stabilizes RpoS, and CsgD itself [35,42]. In addition, *csgD* expression and CsgD activity can be influenced by the bacterial secondary messenger, cyclic-di-GMP (c-di-GMP) [44]. We wanted to investigate how these additional regulatory components influenced metabolic control of the *S. typhimurium* biofilm regulatory network. Strains were grown at 28 °C or 37 °C on 1% tryptone agar, with different components added to the media. To modulate intracellular c-di-GMP levels, strains were transformed with plasmids over-expressing *stm1987*, encoding a DGC enzyme that generates c-di-GMP, or *yhjH*, encoding a PDE enzyme that breaks down c-di-GMP. To analyze the proposed feed-forward loop, we utilized a plasmid over-expressing RpoS and measured gene expression in Δ*rpoS* and Δ*iraP* strains. Each strain was transformed with a luciferase reporter plasmid so that we could visualize *csgBAC* expression.

The vector-only *S*. Typhimurium 14028 control strain displayed robust light production at 28 °C, with faint *csgBAC* signals also observed in the presence of 25mM salt (Figure 7, vector). Over-expression of *rpoS* appeared to elevate *csgBAC* expression under most conditions, including in the presence of salt and at 37 °C (Figure 7, *rpoS*). The importance of RpoS was emphasized in that the Δ*rpoS* strain had no visible *csgBAC* expression under all tested conditions, unless it was co-transformed with pACYC/*rpoS* (Figure 7; 28 °C Δ*rpoS*). A strong stimulatory effect was also caused by over-expression of *stm1987*, which allowed for robust *csgBAC* expression and biofilm colony morphology under most conditions (Figure 7, *stm1987*). The strain transformed with pBR322/*stm1987* was the only one to have detectable *csgBAC* expression at 37 °C in the presence of 25 mM salt (Figure 7). This indicated that elevated levels of c-di-GMP may be enough to overcome temperature-based repression of *csgBAC*. Emphasizing the importance of c-di-GMP, the expression of *yhjH* was sufficient to abolish *csgBAC* expression at 28 °C (Figure 7, *yhjH*), as well as in all other tested conditions. In contrast, deletion of *iraP* appeared to have little effect on *csgBAC* expression, with only a mild reduction observed at 28 °C (Figure 7, *iraP*). Finally, the presence of glucose in the media abolished *csgBAC* expression in all strain and plasmid combinations (Figure 7, 0.2% Glc). This experiment indicated that increased levels of RpoS and c-di-GMP could partially overcome some *csgBAC* repression, and that glucose was perhaps the most powerful metabolic signal feeding into the *S. typhimurium*
*csgD* regulatory network.

## 4. Discussion

Biofilm formation is subject to tight and complex regulation through transcription factor CsgD. In *S*. Typhimurium, the intergenic region between divergent *csgDEFG* and *csgBAC* operons is among the longest non-coding region with 582 bp, which allows for a highly sophisticated signaling network. CsgD expression is regulated at the transcriptional, post transcriptional, translational and post translation level, in response to a variety of external and internal signals [8]. In this study we show that once activated, the CsgD biofilm network responds differently to metabolic inputs.

The ability of *S. enterica* strains to form biofilms is thought to be critical for the success of *Salmonella* as pathogens, particularly for gastroenteritis-causing strains [14]. With bistability of CsgD synthesis resulting in distinct cell types—multicellular aggregates associated with persistence (CsgD-ON), and single cells associated with virulence (CsgD-OFF) [42]—there is a need to have a flexible and dynamic response. We speculated that this phenotypic heterogeneity was a form of bet-hedging. A bet-hedging strategy ensures that at least one group of cells will be more adapted for a specific set of conditions that is encountered [61]. For some bacterial processes, such as sporulation, the advantage of the sporulating cell is obvious; however, for the non-sporulating cells, the advantage lies in being capable of more rapid growth when an influx of new nutrients occurs [62]. For *Salmonella*, there is a lot of energy devoted to generating the polymers associated with biofilm aggregates [26,63]; in the virulent, single cell group, synthesis of the type three secretion apparatus also requires a significant outlay of energy [64]. This type of population split makes the most sense in response to the unpredictability of transmission [65] or perhaps for modulating host–pathogen interactions, as observed for *Vibrio cholerae* [66]. We analyzed regulation before *csgD* activation, which has been tested before in *S. typhimurium* and *E. coli* and generally had the expected results, and compared this to regulation after *csgD* activation, which to our knowledge has not been tested before. We observed that *csgD* transcription and activation of downstream biofilm components was no longer repressed by increased osmolarity, and that the response to nutrient addition was also different, either as individual amino acids or a set of pooled amino acids. In contrast, the addition of glucose and temperatures above 32 °C rapidly repressed *csgD*, *csgB* and *adrA* expression even after induction. We approached these experiments from the point of view of biofilm formation as a developmental process [67,68], and our results show that CsgD biofilm formation is reversible, but can also be viewed as irreversible, depending on the signal. Our results, therefore, suggest the existence of a regulatory hierarchy among external signals that regulate biofilm formation. 

For osmolarity, it has been well established that the optimal conditions for *csgD* expression and rdar biofilm formation in vitro include low osmolarity [51,52,69]. Key transcription factors have been identified (i.e., OmpR, CpxR, H-NS, MlrA and others) and binding within the *csgD* promoter region has been characterized [29,36,37,70,71]. Yet, there are still some intriguing aspects; for example, *S. enterica* biofilm cells produce high levels of osmoprotectants even when growing in low osmolarity conditions [26]. To explain the accumulation of osmoprotectants, we hypothesized that there could be high osmolarity microenvironments created within biofilms due to nutrient and ion trapping by the extracellular matrix [25,72]. The presence of hyperosmolar environments was recently observed with *E. coli* biofilms [73]. Our experiments show that once CsgD has activated downstream target genes (i.e., *csgBAC* (curli) and *adrA* (cellulose), transcription of all units becomes unresponsive to increases in osmolarity. This was specific to the CpxR/A two-component system in high salt conditions and by activating CpxR in ways that are not expected to significantly a change in osmolarity (i.e., metal stress) [55]. There has been some recent controversy about the role of CpxR in surface sensing or adhesion [74], but it is a well-established regulator of *csgD* [75]. The osmolarity effect was also general, as similar gene expression patterns were observed after the addition of sucrose, which was shown to repress *csgD* transcription in *E. coli* by acting through H-NS [36]. In our experiments, the CpxR/A system was not activated by the addition of sucrose, therefore, we assume that the same H-NS-mediated signaling occurs in *Salmonella*. To explain the results with *csgD*, it is possible that the presence of osmoprotectants produced early on during biofilm formation could mute the signaling effects associated with high external osmolarity [76]. Although we have shown that several osmoprotectant-associated genes are produced in time with *csgBAC* and *adrA* [26], we do not know the detailed time course for the appearance of the molecules themselves. The biological relevance for a lack of response to increased osmolarity is not clear, however, a recent paper described a real-world scenario where such a characteristic could be favored. Grinberg et al. 2019 [77] demonstrated that bacterial aggregates have enhanced survival on the surfaces of leaves in microdroplets that are not visible to the naked eye. As liquid evaporates from the leaf surfaces, solutes become concentrated and the microdroplets become hyperosmolar solutions. One could envision *S. enterica* biofilm aggregates surviving well in this scenario due to their stress-resistance adaptations and the altered *csgD* regulatory program identified here. We hypothesize that these microdroplets represent an environment where biofilms, and presumably biofilm-forming strains, would be favored over individual cells that do not aggregate together or strains that do not form biofilms.

Nutrient limitation was one of the first activating signals identified for *csgD* transcription [30,51]. In 1% tryptone or lysogeny broth, which are predominantly comprised of amino acids [60], *csgD* transcription occurs when cell density increases and cells start to run out of nutrients [8,63]. While this was initially attributed to phosphate and nitrogen depletion [30], we tested if supplementation with additional amino acids would delay or prevent activation of *csgD* transcription. When amino acids were added together (i.e., casamino acids), the transcription of *csgD* and downstream biofilm genes was delayed for almost the entire 70-h growth period, well after high cell densities were reached. When CAA were added during growth, *csgD* expression was shifted down in a dose-dependent manner. This showed that after induction, *csgD* expression was still responsive to negative regulation by CAA. The dose response could represent a subpopulation of *S. typhimurium* cells that retain metabolic flexibility [78] and are able to shift their metabolism away from biofilm formation. Based on the results with CAA, we predicted that individual amino acids might also have a repressive effect on biofilm formation. We measured the expression of the curli biosynthesis operon (i.e., *csgBAC*), a direct target of CsgD. Only Asn, Pro and Arg reduced *csgB* expression when added before growth, while Ile, Val, Gln, Met, Ala, Thr and Gly all increased expression. This indicated that the repression observed with CAA was the cumulative effect of the individual amino acids, as recently observed [79]. When added during growth, Leu, Arg, His, Val, Pro, Ala, Gln and Thr increased *csgB* expression, and no single amino acid decreased expression. This again showed that the CsgD biofilm network responds differently once it is activated. The production of sugars from gluconeogenesis is important for biofilm formation, as *S*. Typhimurium strains with mutations of *pckA* and *ppsA* are unable to form biofilms [26]. PckA and PpsA are important gluconeogenic enzymes required for the synthesis of phosphoenolpyruvate (PEP). Pck catalyzes the conversion of oxaloacetate to PEP [80], while Pps catalyzes the conversion of pyruvate to PEP. Ala, Gly and Thr are gluconeogenic amino acids that enter the gluconeogenic pathway through pyruvate [81]. In support of this, Ala and Thr increased *csgB* expression when introduced before and during growth. Gly also increased *csgB* expression when added before and during growth, but the change was not statistically significant. CsgD was shown to directly stimulate Gly biosynthesis during *E. coli* biofilm formation [82], presumably to ensure there is enough Gly supply to produce large quantities of the major curli subunit, CsgA (i.e., 16% Gly residues). Increased *csgB* expression in the presence of Ala, Gly and Thr is consistent with their conversion to pyruvate contributing to gluconeogenesis. For the aromatic amino acids, due to solubility and concentration problems, we only tested Phe, which had no significant effect on *csgB* expression. This was unfortunate since *S. enterica* strains defective in aromatic amino acid biosynthesis are unable to form biofilms [83], and tryptophan has been shown to have an important role in *S*. Typhimurium biofilms [84]. Tryptophan was also not present in CAA, as it is destroyed during the acid hydrolysis process [85]. More research is needed to understand the impact of individual amino acids on *csgD* expression.

Glucose was the most powerful external signal tested in our experiments. Under all growth conditions, the presence of exogenous glucose completely repressed the transcription of *csgD*, *csgB* and *adrA*. Expression of *csgD* was repressed in the presence of glucose even when *rpoS* was over-expressed from a plasmid or when levels of c-di-GMP were enhanced due to STM1987 activity. In the initial paper on carbon source foraging [86], the presence of glucose had a streamlining effect on the metabolism of *E. coli* when compared with growth on lower-quality carbon sources. This study was a genome-wide illustration of carbon catabolite repression [87], where growth on optimal carbon sources occurs first and genes for the metabolism of other carbon sources are repressed, usually acting through cyclic AMP (cAMP) and cAMP receptor proteins (CRP). Glucose had a repressive effect on biofilm formation in both *S*. Typhimurium and *E. coli* [12,59,79,88], however how the regulation is mediated is reported to be the opposite. High levels of cAMP repress *csgD* transcription in *S*. Typhimurium [79], but stimulate *csgD* transcription in *E. coli* [88]. It was also initially reported in *S*. Typhimurium that cAMP/CRP had no effect on *csgD* transcription [30]. It is hard to believe that the conserved divergent *csgDEFG* and *csgBAC* operons [89], biofilm networks and large intergenic region are capable of having opposite regulation in *S*. Typhimurium and *E. coli*. However, as pointed out by Hufnagel et al. [88], *E. coli* and *S*. Typhimurium have different evolutionary histories, hence could have differing regulatory responses to glucose. Another important aspect of cAMP/CRP regulation and glucose metabolism pertains to the quality of nitrogen source available [90], making this complex regulatory network in need of further study. It should be noted that the repressive effect of glucose did not change according to whether *csgD* transcription was activated or not, which was in contrast to the other nutritional signals that we tested.

The effects of temperature and c-di-GMP on *csgD* transcription were also evaluated. Temperature was one of the first conditions identified to regulate biofilm formation [50,52]. Activation at temperatures below 30 °C is known to represent RpoS-dependent transcription of *csgD*. *S. enterica* strains with *csgD* promoter mutations can alleviate temperature-based repression by shifting transcription to be RpoD-dependent [51,52]. This may be a way for natural *rpoS* mutant strains to retain the ability to form biofilms, as there are always a few isolates within natural collections that display temperature-independent biofilm formation [12,21,50]. Temperature was able to shut off the biofilm network even after *csgD* was activated, proving that it is also a strong regulatory signal. *S. typhimurium* biofilm colonies were only formed at 37 °C if c-di-GMP levels were enhanced by *stm1987* overexpression, with partial restoration if *rpoS* was overexpressed. Although these conditions are somewhat artificial, the c-di-GMP regulatory principles could be an important observation. We recently discovered that curli can be synthesized by *S. typhimurium* during murine infections, with *csgD* transcription activated at 37 °C in vivo [91]. It is also of note that iron limitation [52] and exposure to bile [92] can alleviate temperature-based repression of *csgD* transcription. Finally, expression of the c-di-GMP-degrading enzyme, YhjH (or STM3611), was enough to repress *csgD* expression in all tested conditions, which is similar to previous observations [93,94].

## 5. Conclusions

We have started to dissect the external signal hierarchy that regulates *csgD* transcription and CsgD-mediated biofilm formation in *S. enterica*. Most significantly, we identified differences in the regulatory responses based on whether or not *csgD* was activated before being exposed to a signal. These findings are summarized in Figure 8A,B. We hypothesize that the differences upon activation are related to the bistable expression of CsgD [40,42], similar to dedicated processes in other bacterial species. Even seemingly well-understood processes, such as diauxie—the switching of *E. coli* growth between two carbon sources—is subject to heterogeneity, as one sub-population of cells ceases growth once glucose has been exhausted, while the other subpopulation begins to grow on the second carbon source without delay [78]. The diauxic behavior was originally interpreted as the whole population of cells stopping growth during a transition period before starting growth on the second carbon source [95]. We hypothesize that many of the *csgD* regulatory elements that we have examined here are consistent between *S*. Typhimurium and *E. coli* [41], with some notable differences. With respect to phenotypic heterogeneity, we may only fully understand biofilm regulation once we are able to examine the fate of individual cells [27].

## Figures and Tables

**Figure 1 microorganisms-08-00964-f001:**
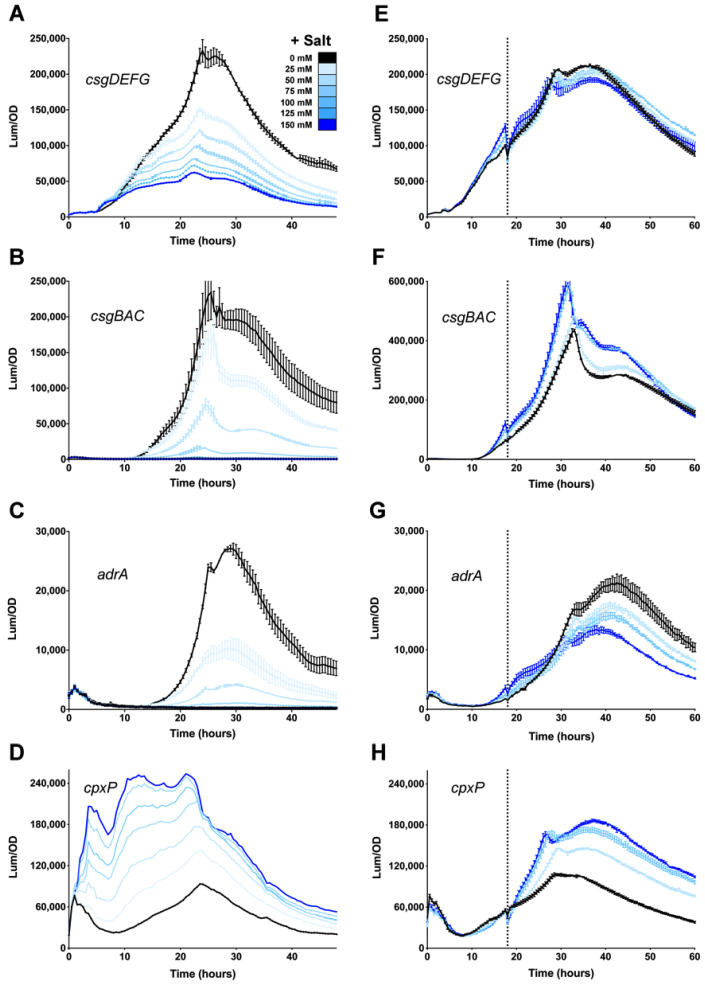
Response of the *Salmonella csgD* regulatory network to changes in osmolarity. *csgDEFG* (**A**,**E**), *csgBAC* (**B**,**F**), *adrA* (**C**,**G**), and *cpxP* (**D**,**H**) expression was measured in *S. typhimurium* 14028 during growth at 28 °C in media premixed with 25, 50, 75, 100, 125 or 150 mM salt (A–D) or with 50, 100, or 150 mM salt added during growth (E–H; vertical line shows the time of addition at 18 h). For each graph, luminescence (light counts per second) divided by the optical density at 600 nm (Lum/OD) was plotted as a function of time with each curve representing a single growth condition. The mean and standard deviations are plotted from experiments performed in triplicate (**A**–**C**, **E**–**H**) or from a single representative experiment (**D**).

**Figure 2 microorganisms-08-00964-f002:**
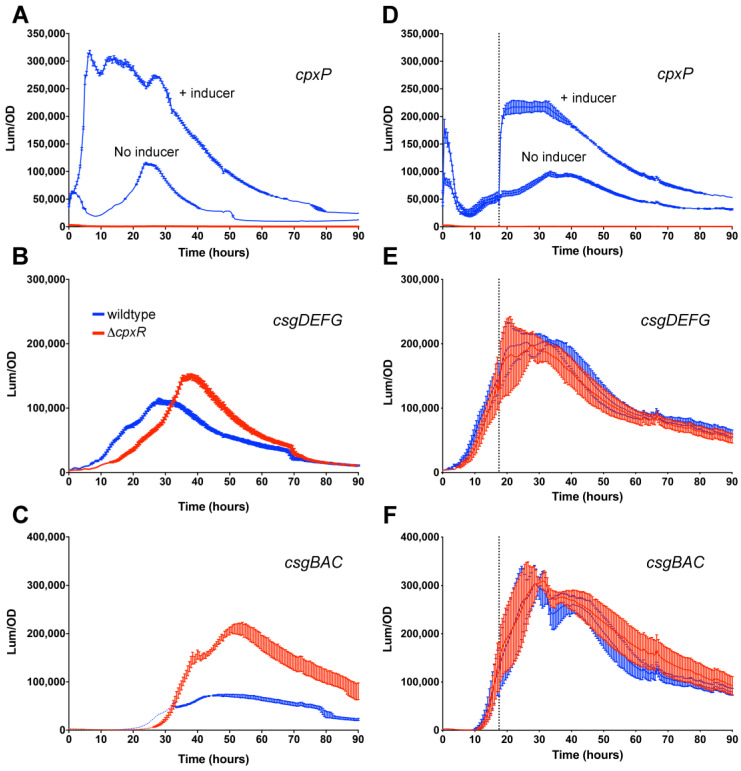
The Cpx system has no repressive effect on *csgD* transcription once the biofilm network is activated. Expression of *cpxP* (**A**,**D**), *csgDEFG* (**B**,**E**), and *csgBAC* (**C**,**F**) operons was measured during growth of *S. typhimurium* 14028 wild-type (blue) or Δ*cpxR* strains (red) at 28 °C in media supplemented with 1.0 mM CuCl_2_ (+ inducer) added at the beginning of growth (**A**–**C**) or added after 18 h of growth (**D**–**F**; the vertical, dotted line represents the time of addition). For each graph, luminescence divided by the optical density at 600 nm (Lum/OD) was plotted as a function of time and each curve represents a single growth condition. The mean and standard deviations are plotted from three biological replicate experiments measured in triplicate.

**Figure 3 microorganisms-08-00964-f003:**
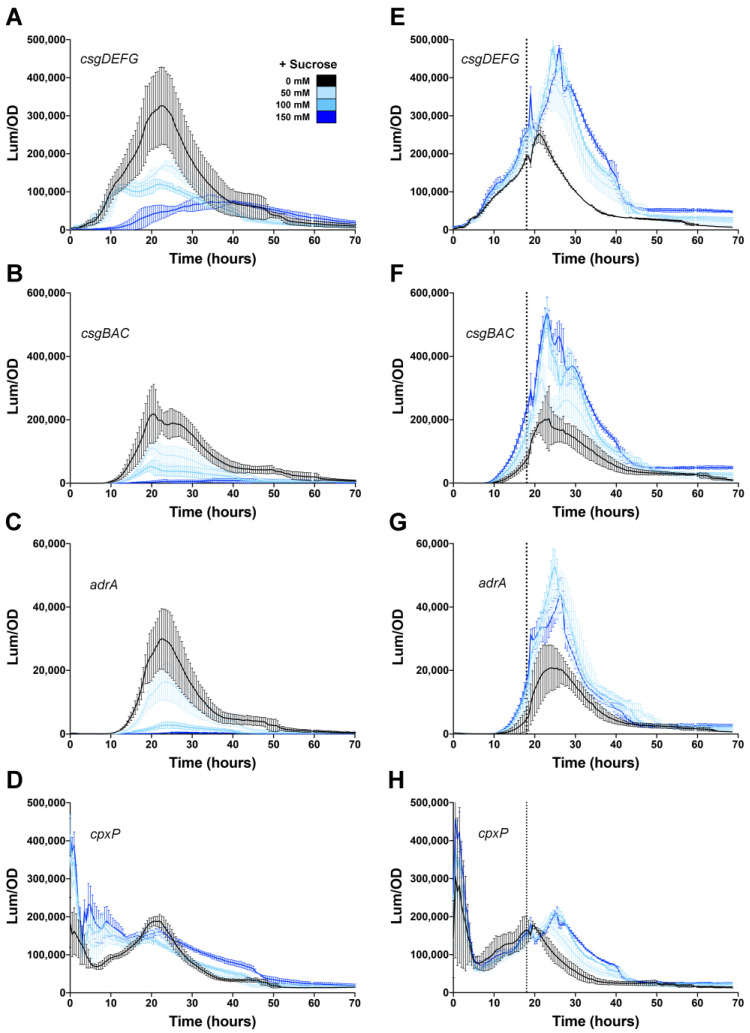
Effect of sucrose addition on the *Salmonella csgD* regulatory network. Expression of *csgDEFG* (**A**,**E**), *csgBAC* (**B**,**F**), *adrA* (**C**,**G**), and *cpxP* (**D**,**H**) operons was measured during growth of *S. typhimurium* 14028 at 28 °C in media premixed with 50, 100 or 150 mM sucrose (**A**–**D**) or with sucrose added during growth (**E**–**H**; vertical line represents the time of addition at 18 h). For each graph, luminescence (light counts per second) divided by the optical density at 600 nm (Lum/OD) is plotted as a function of time and each curve represents a single growth condition. The mean and standard deviations are plotted from three biological replicate experiments measured in triplicate.

**Figure 4 microorganisms-08-00964-f004:**
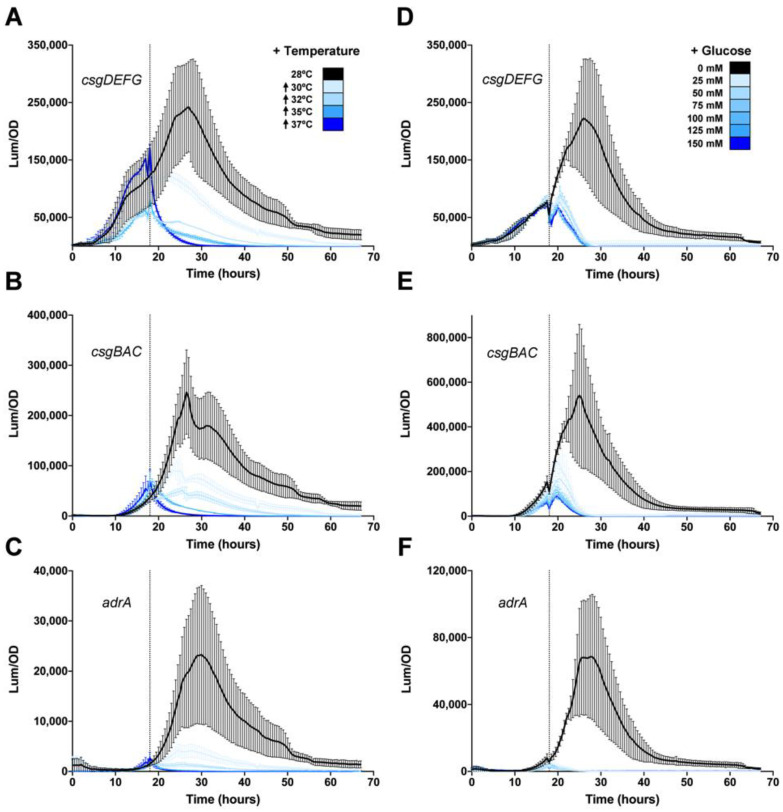
The *csgD* biofilm network in *Salmonella* is repressed by the addition of glucose or an increase in growth temperature. Expression of *csgDEFG* (**A**,**D**), *csgBAC* (**B**,**E**), and *adrA* (**C**,**F**) was measured during growth of *S. typhimurium* 14028 at 28 °C for 18 h prior to temperature shift (**A**–**C**) or the addition of 25, 50, 75, 100, 125, or 150 mM glucose (**D**–**F**). The vertical dotted line represents the time of temperature shift or glucose addition. For each graph, luminescence divided by the optical density at 600 nm (Lum/OD) is plotted as a function of time and each curve represents a single growth condition. The mean and standard deviations are plotted from three biological replicate experiments measured in triplicate.

**Figure 5 microorganisms-08-00964-f005:**
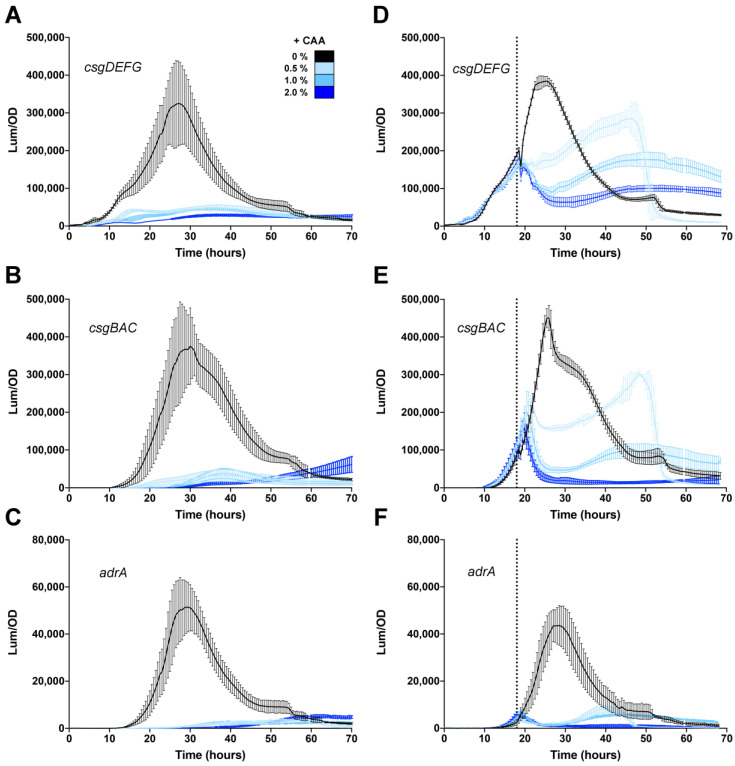
The *csgD* biofilm regulatory network in *Salmonella* is repressed by the addition of amino acids. Expression of *csgDEFG* (**A**,**D**), *csgBAC* (**B**,**E**), and *adrA* (**C**,**F**) was measured during growth of *S. typhimurium* 14028 at 28 °C in media premixed with 0.5%, 1.0% or 2.0% casamino acids (**A**–**C**) or in media where casamino acids were added during growth (D, E, F; the dotted line represents the time of addition at 18 h). For each graph, luminescence (light counts per second) divided by the optical density at 600 nm (Lum/OD) is plotted as a function of time and each curve represents a single growth condition. The mean and standard deviations are plotted from three biological replicate experiments measured in triplicate.

**Figure 6 microorganisms-08-00964-f006:**
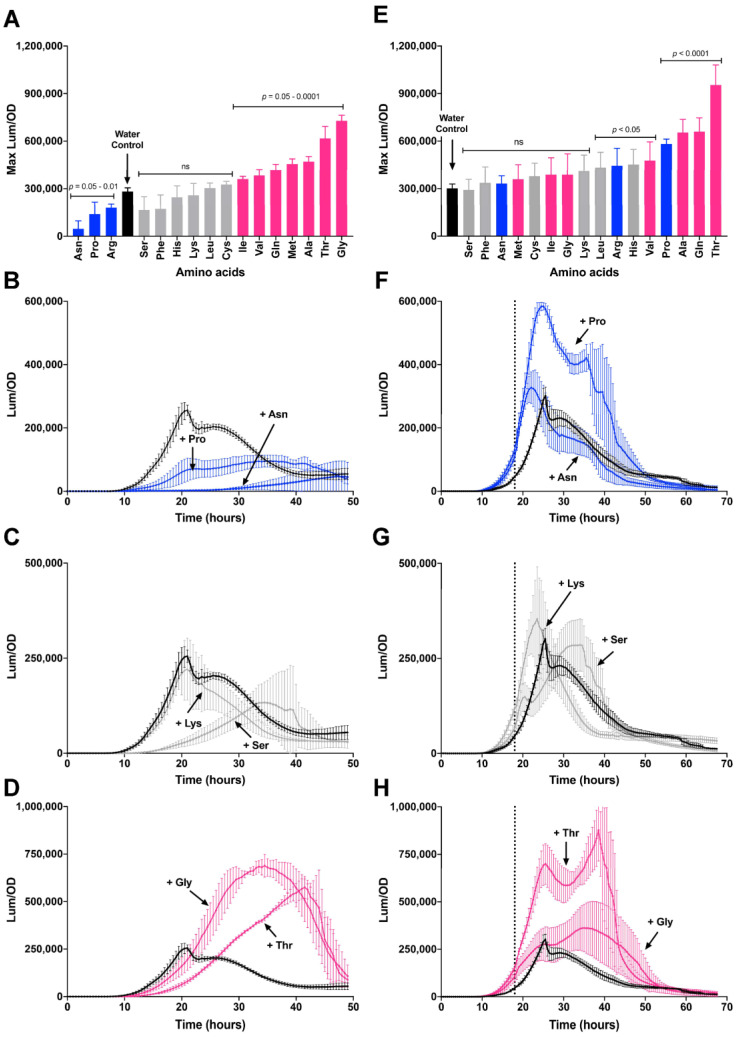
Individual amino acids have differing effects on the *csgD* biofilm regulatory network in *S. typhimurium* 14028. Maximum expression of the *csgBAC* operon (curli production) was recorded during growth of *S. typhimurium* 14028 at 28 °C in media premixed with 15 mM of individual amino acids (**A**) or in media where the amino acids were added after 18 h of growth. The maximum Lum/OD values after addition of each amino acid were statistically compared to a water control and amino acids were determined to have a repressive (blue), neutral (grey) or stimulatory effect (purple) on *csgB* expression (**A**). This color scheme was used to represent the same amino acids when they were added after 18 h of growth (**E**). Lum/OD values were plotted as a function of time corresponding to selected amino acids premixed into the media (**B**–**D**) or added at 18 h of growth (**F**–**H**; the dotted line represents the time of addition). For each curve, the mean and standard deviations are plotted from three biological replicate experiments measured in triplicate.

**Figure 7 microorganisms-08-00964-f007:**
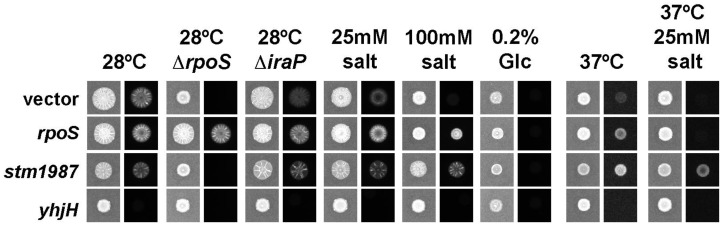
Visualization of *S. typhimurium* curli expression in response to changing growth conditions. *S. typhimurium* 14028 wild-type, Δ*rpoS* or Δ*iraP* reporter strains containing a *csgBAC* promoter–luciferase fusion were transformed with pBR322 (vector), pACYC/rpos (*rpoS*), pBR322/stm1987 (*stm1987*) or pBR322/yhjH (*yhjH*) plasmids. Cells were inoculated onto T agar or T agar supplemented with 0.2% glucose, 25 mM or 100 mM NaCl and grown at 28 °C or 37ºC. Colony morphology (left column) and luminescence (right column) was recorded after 48 h growth. Control strains containing pACYC were also tested, but the *csgBAC* expression profiles were similar to strains transformed with pBR322; therefore, only the pBR322 pictures are shown.

**Figure 8 microorganisms-08-00964-f008:**
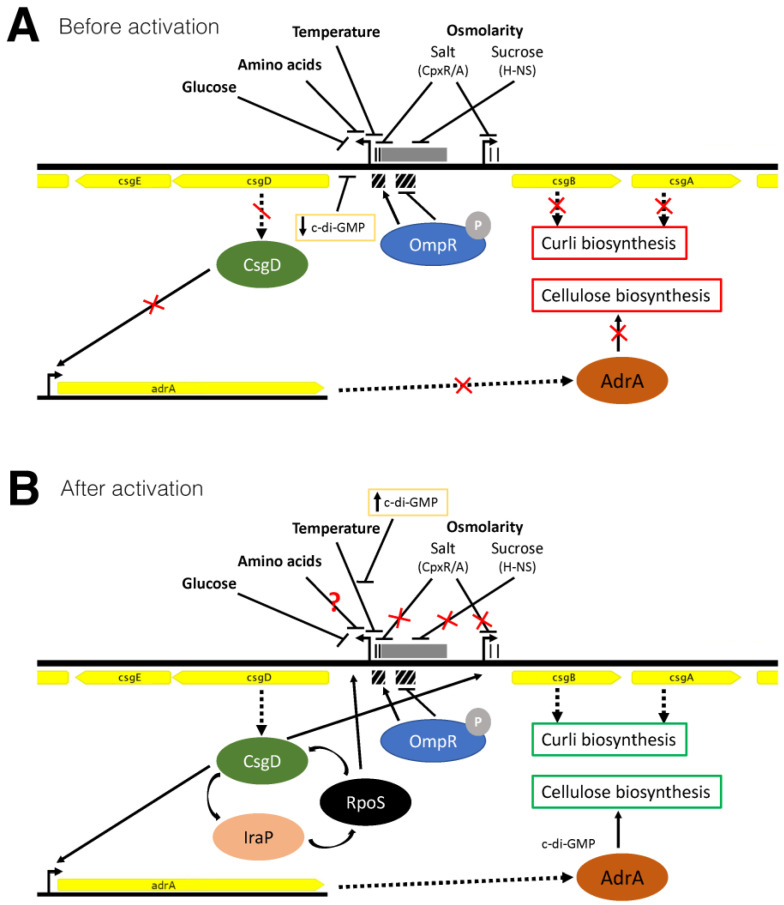
Graphical illustration of the CsgD regulatory principles identified in this manuscript. The divergent *csg* operons are shown (without *csgFG* and *csgC*) with the intergenic region highlighted by transcription factor binding sites that have been experimentally verified in *Salmonella* (CpxR—black bars; H-NS—grey box; OmpR—hatched boxes). Phosphorylated OmpR binds the proximal, high affinity site under conditions of low osmolarity, which activates *csgD* transcription, and binds the distal, low affinity sites under conditions of high osmolarity, which represses *csgD* transcription [38]. The different regulatory elements that we have tested are shown: glucose; amino acids; growth temperature; and osmolarity, with sodium chloride, which is known to act via the CpxR/A system [37], and sucrose, which is known to act via H-NS [36]. The *adrA* gene encodes a diguanylate cyclase, which produces cyclic-di-GMP and allosterically activates cellulose production. (**A**) Glucose (>25 mM), amino acids (>0.5% casamino acids), temperature (>32 °C), salt and sucrose (>25 mM) caused a reduction in *csgD* transcription and blocked transcription of *csgBAC* and *adrA*, preventing curli and cellulose biosynthesis. The effect of reduced c-di-GMP was tested by overexpression of the YhjH phosphodiesterase. The addition of individual amino acids was variable, with three leading to reduced *csgD* transcription (Asn, Pro, Arg), and seven leading to increased *csgD* transcription (Ile, Val, Gln, Met, Ala, Thr, Gly). (**B**) When the same regulatory components were tested after 18 h of growth, the effects were different. We assume that by this time point, the CsgD-IraP-RpoS feed-forward loop [35] is activated, although deletion of *iraP* in our experiments had little effect. The addition of salt and sucrose had no effect on *csgD* transcription, and casamino acids were not as repressive. The effect of increased c-di-GMP was tested by overexpression of the diguanylate cyclase STM1987, which was able to relieve temperature-based repression of *csgD* transcription. The response to individual amino acids was again variable, however, none caused a reduction in *csgD* transcription and eight were stimulatory (Leu, Arg, His, Val, Pro, Ala, Gln, Thr). The question mark signifies that we do not fully understand the regulatory effects of individual amino acids.

**Table 1 microorganisms-08-00964-t001:** Strains and plasmids used in this study.

Strains or Plasmids	Genotype	Reference
Strains		
*Salmonella enterica* subsp*. enterica* serovar Typhimurium		
14028	Wild-type strain	ATCC
14028 ∆*cpxR*	Deletion of *cpxR*	This study
14028 ∆*iraP*	Deletion of *iraP*	This study
14028 ∆*rpoS*	Deletion of *rpoS*	[12]
Plasmids		
pCS26, pU220	Bacterial luciferase	[47]
pCS26-*stm1987*::*luxCDABE*	*stm1987* promoter	This study
pU220-*cpxP*::*luxCDABE*	*cpxP* promoter	This study
pU220-*csgD*::*luxCDABE*	*csgDEFG* promoter	[12]
pCS26-*csgB*::*luxCDABE*	*csgBAC* promoter	[12]
pCS26-*adrA*::*luxCDABE*	*adrA* promoter	[12]
pBR322		
pBR322/*stm1987*	*stm1987*^14028	This study
pBR322/*yhjH*	*yhjH*^14028	This study
pACYC184		
pACYC/*rpoS*	*rpoS*^14028	[48]

**Table 2 microorganisms-08-00964-t002:** Oligonucleotides used in this study.

Primer	Sequence (5′–3′) ^a^	Purpose
cpxRko sense	AAGATGCGCGCGGTTAAACTTCCTATCATGAAGCGGAAACCATCAGATAGGTGTAGGCTGGAGCTGCTTC	To amplify *cat* gene product from pKD3 to generate ∆*cpxR* strain by lambda red recombination
cpxRko antisense	CCTGTTAGTTGATGATGACCGAGAGCTGACTTCCCTGTTAAAAGAGCTCCCCTCCTTAGTTCCTATTCCG	
cpxR ver F	CCAGCATTAGCACCAGCGCC	To confirm the deletion of *cpxR* from *S. typhimurium* 14028
cpxR ver R	TCTGCCTCGGAGGTACGTAAACA	
cpxR1	GCCCTCGAGGTAACTTTGCGCATCGCTTG	To amplify the *cpxR* and *cpxP* promoter regions from *S. typhimurium* 14028
cpxR2	GCCGGATCCTTCATTGTTTACGTACCTCCG	
iraPko sense	GGCAGTGGTTCTTCATAGTGATAACGTCACCCTGGAACTAATAAGGAAATGTGTAGGCTGTAGCTGCTTC	To amplify *cat* gene product from pKD3 to generate ∆*iraP* strain by lambda red recombination
iraPko antisense	TGTTATTTCATAAAAGTAACGTTATAACAACTGTGTTGTTTTAAATACGACCTCCTTAGTTCCTATTCCG	
iraPko-detect1 iraPko-detect2	CAAAAAGCGAAAGGCCAATA TAGCACCATCCTTTTGTCAG	To confirm the deletion of *iraP* from *S. typhimurium* 14028
STM14_2408for1	GATCCTCGAGAAATTCGCGGTGTTTCGCAC	To amplify the *stm1987* promoter region from *S. typhimurium* 14028
STM14_2408rev2	GATCGGATCCCTAACAGTGTTTCGTGCGGC	
STM1987forEco	GATCGAATTCAAACGGTGTTTCGCAC	To amplify *stm1987* with native promoter region from *S. typhimurium* 14028
STM1987revAatII	GATCGACGTCGGACTATTTCTTTTCCCGCT	
yhjHforEco	GATCGAATTCTTGACAAGTTTCGGGGGCTG	To amplify *yhjH* with native promoter region from *S. typhimurium* 14028
yhjHrevAatll	GATCGACGTCGTATTACGGGAACAGTCTGG	
pZE05	CCAGCTGGCAATTCCGA	Used to verify promoter fusions to *luxCDABE*
pZE06	AATCATCACTTTCGGGAA	

^a^ Nucleotide sequences corresponding to restriction enzyme sites are underlined.

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
