# Peer review of "Metabolic Activation of CsgD in the Regulation of Salmonella Biofilms"

_microorganisms, 2020, doi:10.3390/microorganisms8070964_

Round 1

Reviewer 1 Report

The regulation and function of csgD gene in Salmonella typhimurium have been widely studied, especially in biofilm formation. In this manuscript, the regulation of csgD and its activity through activation of csgBAC and adrA in response to different environmental signals including temperature, osmolarity, and nutrients were studied. However, environmental conditions such as temperature, oxygen tension, starvation, osmolarity, iron and pH had been identified to regulate csgD expression (Sukupolvi et al., 1997; Römling et al., 1998; Gerstel and Römling, 2001; Prigent‐Combaret et al., 2001). The novelty and significance of the current study remain vague and should be highlighted. An informative graphical illustration is recommended to show the major regulation of csgD concluded from this study.

Also, environmental signals including temperature, osmolarity, and nutrients were selected, the reasons should be added at least in the introduction. Are these based on the environmental conditions frequently appeared during food processing or in other host species?

In line 27, “Temperature mediated regulation of csgD on agar was altered by intracellular levels of RpoS and cyclic-di-GMP.” RpoS had been reported to contribute to the formation of bacterial VBNC state. The culturability and viability of cells may differ. Have you tired other biofilm models? As VBNC cells may be omitted in agar based methods. Concerning c-di-GMP, detailed analysis of c-di-GMP mediated regulation of csgD expression in Salmonella typhimurium had been reported (Ahmad et al. BMC Microbiology 2017), any further findings than this paper?

The formatting should be improved. For example, in line 546, S. Typhimurium and E. coli should be italic and the first letter of Typhimurium should not be capitalized.

Author Response

Comment from Reviewer 1: The regulation and function of csgD gene in Salmonella typhimurium have been widely studied, especially in biofilm formation. In this manuscript, the regulation of csgD and its activity through activation of csgBAC and adrA in response to different environmental signals including temperature, osmolarity, and nutrients were studied. However, environmental conditions such as temperature, oxygen tension, starvation, osmolarity, iron and pH had been identified to regulate csgD expression (Sukupolvi et al., 1997; Römling et al., 1998; Gerstel and Römling, 2001; Prigent‐Combaret et al., 2001). The novelty and significance of the current study remain vague and should be highlighted. An informative graphical illustration is recommended to show the major regulation of csgD concluded from this study.

Our Response: We agree with the reviewer about this point. There has been a lot of work done studying csgD regulation. However, it is now accepted that CsgD is produced in a bistable manner and that the overall biofilm population splits into differentiated cell types. Our study arose because one trainee added different regulatory factors to growing S. Typhimurium cultures and noticed that the regulatory pattern was different than when the regulatory factors were added before growth occurs. In each of the papers cited by the Reviewer, the regulatory factors were incorporated before growth. As we have shown, when we repeat what others have done, the results are similar or are the same. When we add or change regulatory factors to growing cultures (i.e., after csgD is activated) the responses are quite different. To our knowledge this has not been reported before. When reading about other bistable genetic systems, the principle of irreversibility has been noticed before, especially for sporulation and other survival-related outcomes. We wanted to test if S. Typhimurium biofilm formation was the same, and by extension to E. coli. I think the finer points of the regulation we have identified will make the most sense when we are able to examine the fate of individual cells (i.e., single-cell microscopy). As suggested by the reviewer, we have added a graphical illustration as Figure 8 at the end of the Discussion.

Comment from Reviewer 1: Also, environmental signals including temperature, osmolarity, and nutrients were selected, the reasons should be added at least in the introduction. Are these based on the environmental conditions frequently appeared during food processing or in other host species?

Our Response: The following statement has been included in the introduction. The environmental signals that we tested were selected because their effects on csgD transcription have been well established. In addition, we believe that these conditions could be frequently encountered during food processing and in both host and non-host environments. We have added a statement to this effect in the last paragraph of the Introduction

Comment from Reviewer 1: In line 27, “Temperature mediated regulation of csgD on agar was altered by intracellular levels of RpoS and cyclic-di-GMP.” RpoS had been reported to contribute to the formation of bacterial VBNC state. The culturability and viability of cells may differ. Have you tried other biofilm models? As VBNC cells maybe omitted in agar-based methods.

Our Response: We analyzed RpoS to probe into the proposed feed-forward loop between RpoS-CsgD and IraP. We wanted to see if increased RpoS synthesis would over-ride some of the regulatory inputs into csgD transcription. We did not investigate other biofilm models so we do not know the impact on the VBNC state. We are interested in the VBNC state (see Apel et al., 2009) but experiments in this area are difficult without a clear path to achieve this state. We would like to pursue this further in the future.

Concerning c-di-GMP, detailed analysis of c-di-GMP mediated regulation of csgD expression in Salmonella typhimurium had been reported (Ahmad et al. BMC Microbiology 2017), any further findings than this paper?

Our Response: Our goal was not to repeat what has been reported by others. We used STM1987 and YhjH expressed from plasmids as a way to artificially increase or decrease intracellular cyclic-di-GMP levels, and see what influence this would have on csgD transcription and the reversibility of the system. In general, we confirmed what others have found, but originally we thought that high levels of c-di-GMP might make the csgD network unresponsive to external inputs. This was not the case.

Comment from Reviewer 1: The formatting should be improved. For example, in line 546, S. Typhimurium and E. coli should be italic and the first letter of Typhimurium should not be capitalized.

Our Response: We have carefully gone through the manuscript and checked all of the text formatting. We hope to have caught all typos. The proper nomenclature for Salmonella is Salmonella enterica subspecies enterica serovar Typhimurium, but no one likes to write this every time. For convention, the name is shortened to S. (italicized) and Typhimurium (capitalized and not italicized). Salmonella serovar Typhimurium is also accepted.

Reviewer 2 Report

In the manuscript “Metabolic activation of CsgD in the regulation of Salmonellabiofilms” by Sokaribo et al., the authors analyze the regulation of curli fimbriae and cellulose production by monitoring their transcription and that of the master biofilm regulator CsgD in response to several environmental factors known to influence biofilm formation. They test this before the onset of biofilm induction and when the system has already been activated. Their results positively contribute to the field of Salmonella biofilm regulation. Below are some comments and suggestions that could be addressed in a revised manuscript. 

Major comments:

Comment 1: 

General comment. I’m curious as to why the luminescence readings were represented as Lum/OD. Presumably, once the cells start to attach to the bottom of the plate and form biofilms OD600would not be measured reliably, especially at later time points. This is additionally puzzling as this group has reported only Luminescence (CPS) readings in previous studies, using the same, or similar reporter plasmids (ie. MacKenzie et al 2019. Plos Genetics, MacKenzie et al 2015. I&I, White et al. 2006 JBact.). Furthermore, in the results section, when describing the figures, some values are given as CPS (e.g. line 301).

Comment 2:

General comment. Keep the same scale for luminescence values on the Y-axis of corresponding figure panels throughout the paper (e.g. Fig 1 panels B & F; Fig 2 panels A & D, B & E; C & F; etc…). This would make it easier to visualize the effect of the change in condition, the addition of an inducer/repressor. It is also a more realistic comparison of differences.

Comment 3: 

General comment. In general, it would be worth mentioning why the 18 h time point was chosen for the addition of inducers (or inhibitors). It would seem that at this time point, for most assays, csgDexpression has been activated and csgABCis just starting to activate, but the reason was not mentioned anywhere in the manuscript.

Comment 4: 

Section 3.3 - The authors’ results clearly show that glucose represses curli expression. Nonetheless, the lowest concentration tested was still quite high. 25 mM is higher than the 0.4% concentration normally used for E. colior Salmonellagrowth in minimal media. I would not expect that Salmonellafaces carbon limiting conditions at this concentration, especially in 1% tryptone broth. Did the authors try lower concentrations? 

Comment 5:

Section 3.5 - It would be interesting to see what happens if this experiment is repeated using a mix of the purified amino acids to see if the results obtained from casamino acids can be replicated. This would also rule out the possible effects of small peptides present in the casamino acids mix. 

Minor comments: 

Line 2: Salmonellashould be italicized in the title

Table 2. Consider using consistent nomenclature of primers instead of switching from sense anti-sense, F, rv, etc...

Fig 2. The color code of lines blue for wild type and red for cpxR mutant found in panel B should be moved (or replicated) in panel A. Since the red line there is practically even with the X-axis, I had to look very closely to know that there was a line there at all for cpxR mutant.

Fig 4. In Panel A ,csgDEFGexpression seems to split off in two groups before changing temperature with 28 and 37 °C expressing a significantly higher amount of csgDEFGthan 30, 32, and 35 °C. What could explain this? 

Line 301: Luminescence is reported in CPS in the text and Lum/OD in Figure 5

Lines 345 and 349: There is a € symbol that should not be there.

Discussion section.

It might be beneficial to speculate why the system is no longer responsive to inducers once it has kicked in for some factors like osmolarity, but it is for others like glucose and temperature. Is CsgD bound to the csgBACpromoter? Do you think glucose can reverse this interaction?

Author Response

Reviewer 2:

Comment from Reviewer 2: I’m curious as to why the luminescence readings were represented as Lum/OD. Presumably, once the cells start to attach to the bottom of the plate and form biofilms OD would not be measured reliably, especially at later time points. This is additionally puzzling as this group has reported only Luminescence (CPS) readings in previous studies, using the same, or similar reporter plasmids (ie. MacKenzie et al 2019. Plos Genetics, MacKenzie et al 2015. I&I, White et al. 2006 JBact.). Furthermore, in the results section, when describing the figures, some values are given as CPS (e.g. line 301).

Our response: We agree with the reviewer. In principle, there is not a big difference in plotting luminescence or luminescence over OD, especially for strong promoters like csgD, csgB and adrA. However, over the past several years many scientists on student committee’s have commented on the need for Lum/OD to have some relationship to growth. In our experience, the OD, as measured in 96 well plates, is not influenced heavily when csgD and csgB are activated. We have removed any mention of CPS and Lum/OD values from the main text and instead refer to the fold-change differences.

Comment 2: General comment. Keep the same scale for luminescence values on the Y-axis of corresponding figure panels throughout the paper (e.g. Fig 1 panels B & F; Fig 2 panels A & D, B & E; C& F; etc…). This would make it easier to visualize the effect of the change in condition, the addition of an inducer/repressor. It is also a more realistic comparison of differences.

Our Response: We have changed the scales in each Figure to make them consistent especially when comparing addition before growth to addition during growth. The exception was for Figure 1, panels B and F, because if we changed the scale in B, we would lose clarity for some of the expression curves shown at higher salt conditions. Therefore, we left the y-axis scales for panels B and F to be like they were.

Comment 3: General comment. In general, it would be worth mentioning why the 18 h time point was chosen for the addition of inducers (or inhibitors). It would seem that at this time point, for most assays, csgD expression has been activated and csgABC is just starting to activate, but the reason was not mentioned anywhere in the manuscript.

Our Response: We have inserted a new statement in the result section “At 18 hours of growth, csgD expression level is rapidly increasing and csgBAC and adrA expression are just beginning to increase" and we cite an earlier paper where expression was measured at these precise time points (White et al., 2006).

Comment 4: Section 3.3 - The authors’ results clearly show that glucose represses curli expression. Nonetheless, the lowest concentration tested was still quite high. 25 mM is higher than the 0.4% concentration normally used for E. coli or Salmonella growth in minimal media. I would not expect that Salmonella faces carbon limiting conditions at this concentration, especially in 1% tryptone broth. Did the authors try lower concentrations?

Our Response: Due to COVID, our laboratory access is restricted at the moment. We have performed some preliminary experiments with lower concentrations of glucose, but unfortunately they are not of sufficient quality to include here. No effect on csgD expression was observed below 2 mM glucose, but there was a sharp decrease with 5 and 10 mM glucose added, with csgD expression returning presumably once the available glucose was metabolized. A statement has been added to this effect in the Results section.

Comment 5: Section 3.5 - It would be interesting to see what happens if this experiment is repeated using a mix of the purified amino acids to see if the results obtained from casamino acids can be replicated. This would also rule out the possible effects of small peptides present in the casamino acids mix.

Our Response: A dose dependent decrease of csgB expression and biofilm formation by a mixture of 20 amino acids has been shown by Paytubi et al., 2017 (Reference #79). A mixture of 1x amino acids was shown to reduce csgB expression and biofilm formation to the same level as casamino acids.

Minor comments:

Line 2: Salmonella should be italicized in the title

Our Response: This has been done

Table 2. Consider using consistent nomenclature of primers instead of switching from sense anti-sense, F, rv, etc...

Our Response: We agree with the reviewer, however we find it best to be consistent with how the primers were ordered – to keep primer stocks organized. Whenever possible we try to keep consistent nomenclature, but your point is taken.

Fig 2. The color code of lines blue for wild type and red for cpxR mutant found in panel B should be moved (or replicated) in panel A. Since the red line there is practically even with the X axis, I had to look very closely to know that there was a line there at all for cpxR mutant.

Our Response: We have increased the weight of the red line in Figure 2A so that it is easier to see.

Fig 4. In Panel A, csgDEFG expression seems to split off in two groups before changing temperature with 28 and 37 °C expressing a significantly higher amount of csgDEFG than 30, 32, and 35 °C. What could explain this?

Our Response: Sometimes with lux assays performed on different machines, there is a discrepancy with CPS values. We think this is what has occurred here and don’t believe there is a significant difference in expression. The main point was testing if temperature could override the system once it had been activated.

Line 301: Luminescence is reported in CPS in the text and Lum/OD in Figure 5

Our Response: We have removed this statement in the main text and refer instead to fold change between conditions.

Lines 345 and 349: There is a € symbol that should not be there.

Our Response: The € symbols has been removed

Discussion section.

It might be beneficial to speculate why the system is no longer responsive to inducers once it has kicked in for some factors like osmolarity, but it is for others like glucose and temperature. Is CsgD bound to the csgBAC promoter? Do you think glucose can reverse this interaction?

Our Response: Thank you for this point. We do not know why glucose is such a powerful signal. It could be related to central carbon metabolism, kind of negative feedback where gluconeogenesis is reversed when glucose is present. However, this is pure speculation at the moment. There is also the possibility that CRP can interfere with CsgD binding at the csgB promoter but the conflicting results published so far with E. coli and Salmonella make this confusing.  

Reviewer 3 Report

Minor comments:

English language style required checking in the whole of the manuscript. For example, in the line 396: S. Typhimurium should be in italic.

I recommend the authors to summarize their results in a scheme to make easier their understanding. 

Author Response

Minor comments:

English language style required checking in the whole of the manuscript. For example, in the line 396: S. Typhimurium should be in italic.

Our Response: We have carefully gone through the manuscript and checked all of the text formatting. We hope to have caught all typos. The proper nomenclature for Salmonella is Salmonella enterica subspecies enterica serovar Typhimurium, but no one likes to write this every time. For convention, the name is shortened to S. (italicized) and Typhimurium (capitalized and not italicized). Salmonella serovar Typhimurium is also acceptable.

I recommend the authors to summarize their results in a scheme to make easier their understanding. 

Our Response: We have included a graphical illustration of our findings regarding CsgD regulation as Figure 8 in the revised manuscript